# Context-adaptive policy framework for robust and reactive robotic manipulation via uncertainty-aware imitation learning

*Abstract*— Generating robust and reactive manipulation strategies that can adapt to changing context information is a challenging task in robotics. Over the years, Learning from Demonstration (LfD) has emerged as an intuitive and effective solution for generating reactive policies, particularly by following dynamical-system(DS)-based approaches. However, most state-of-the-art DS-based approaches focus on addressing the stability problem, overlooking the modulation of policies in response to the environment. As a result, they tend to be inflexible with respect to parameterization by task-dependent variables. In this paper, we build on existing work on policy fusion and uncertainty quantification to propose a context-adaptive policy framework that combines task-parameterized, robust and reactive manipulation. For this, we use LfD to acquire a policy that is conditioned on the robot state and task-dependent parameters reflecting the environment. We combine the learned policy with additional uncertainty-aware policies using a Mixture of Experts (MoE) formulation to improve its stabilization and convergence properties. The approach is evaluated on the LASA handwriting dataset and on a real 7-DoF robot in three scenarios: force-conditioned grasping, manipulation of deformable food items and dynamic grasping.

## I. INTRODUCTION

Context-adaptive manipulation is crucial in tasks in which the robot needs to adapt its motion to its state and task-dependent parameters, such as contact forces. Due to the complexity inherent in modeling this context[1] information, the application of conventional planning and control algorithms is restricted. Consequently, recent approaches focus on solving this problem using data-driven machine learning techniques. Especially, the application of LfD approaches in the domain of robotic manipulation has gained popularity in recent years, due to the typically reduced data requirements and the adaptability to new tasks [1]. These LfD approaches usually learn a probabilistic policy $p_{\boldsymbol{\theta}}(\boldsymbol{\xi}|\boldsymbol{s})$ that maps states $\boldsymbol{s}$ to actions $\boldsymbol{\xi}$ based on demonstrated data and can be executed to reproduce the demonstrated behavior at test input $\boldsymbol{s}_*$, i.e. $\boldsymbol{\xi}_* = f_{\boldsymbol{\theta}}(\boldsymbol{s}_*)$, with $\boldsymbol{\theta}$ denoting the policy parameters.

While time-variant policies (e.g., $p_{\boldsymbol{\theta}}(\boldsymbol{\xi}|t)$), including regression approaches such as Gaussian Mixture Regression (GMR) [2], [3], Gaussian Process Regression (GPR) [4], and Locally Weighted Regression (LWR) [5], as well as movement primitive (MP)-based approaches, such as Dynamic MP (DMP) [6], Probabilistic MP (ProMP) [7], and Kernelized MP (KMP) [8], demonstrate efficacy in static tasks, their strict time-dependence impose significant limitations

on spatial robustness and adaptability. Extensions like Task-Parameterized Gaussian Mixture Models (TP-GMM) [3], [9], Task-Parameterized DMP (TP-DMP) [10], [11], and Task-Parameterized ProMP (TP-ProMP) [12], [13] enhance generalization through task-parameterization, but rely on strong assumptions regarding the task-parameters and demonstrate limited adaptability in dynamic environments.

In contrast, DS-based policies, which learn a function from robot state $\boldsymbol{x}$ to its derivative $\dot{\boldsymbol{x}}$, i.e. $p_{\boldsymbol{\theta}}(\dot{\boldsymbol{x}}|\boldsymbol{x})$, enable real-time adaptation through continuous state feedback, offering strong reactivity. However, they often introduce stability issues in case of disturbances, different initial conditions or distribution shifts. Recent advances like diffusion policies [14] or flow-based policies [15] aim to achieve robust behavior by increasing the diversity and coverage of the training distribution, but at the cost of high computational overhead, no formal stability, and limited generalization beyond training distributions. Efforts to enforce stability using *hard* control-theoretic stability constraints [16]–[19] or *soft* guarantees based on policy fusion [20]–[22] improve robustness but typically restrict inputs and outputs to the robot state and its derivative, sacrificing adaptability to the environment. To the best of the authors' knowledge, a unified approach combining real-time adaptation, stability guarantees, and modulation by arbitrary parameters remains unexplored.

In this work, we address this gap by designing a robust DS-based policy framework that can be modulated in response to the environment. For this, we propose an MoE [23] formulation with uncertainty-aware experts and mixing coefficients, where individual expert policies are active in distinct regions of the state space. We employ Gaussian Process Regression (GPR) [24] due its data efficiency to learn an *LfD policy* conditioned on the multidimensional robot state and task-dependent parameters. Moreover, we make use of its analytical uncertainty formulation to derive a *stabilizing policy* that guides the robot back to the demonstrated trajectories, preventing undesired behavior, and design a kernel-based *goal attractor policy* to ensure convergence at the demonstrated goal states. Concretely, the main contributions of this work can be summarized as:

- Incorporating task-dependent parameters explicitly into the learned policy (II-A), in addition to the robot state.
- Extending the stabilizing policy of [21], [22] to include orientation as part of the state (II-B), and introducing a novel kernel-based goal attractor policy (II-C).
- Formulating the problem as an MoE framework with varying, uncertainty-aware mixing coefficients (II-D).

---

[1]In this work, we refer to context as the combination of robot state and task-dependent parameters that reflect the environment.

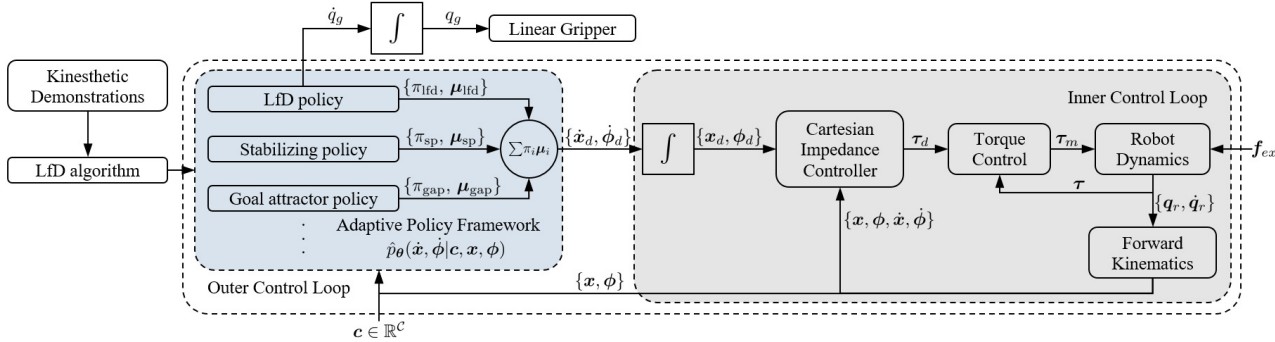

Fig. 1: Schematic overview of the two-layered control loop structure with the context-adaptive policy framework (highlighted in blue) that generates desired end-effector velocities in the outer control loop and the inner control loop (highlighted in gray) that makes the robot follow the desired velocities.

## II. CONTEXT-ADAPTIVE POLICY FRAMEWORK

The aim of the context-adaptive policy framework is to provide a general framework for robust and reactive robotic manipulation within dynamically changing environments. In order to achieve the reactive behavior, we apply a two-layer control loop structure, displayed in Fig. 1, as the basis of our approach. Here, the inner control loop (highlighted in gray), consisting of a low-level controller and the robot's dynamics, makes the robot follow desired linear and angular end-effector velocities $\dot{x}_d$, $\dot{\phi}_d$, which are obtained in the outer control loop by querying a context-adaptive policy $\hat{p}_{\boldsymbol{\theta}}(\dot{x}, \dot{\phi}|c, x, \phi)$. This DS-based policy is conditioned on the end-effector pose, defined by position $x$ and orientation $\phi$, and on low-dimensional task-dependent parameters $c \in \mathbb{R}^{\mathcal{C}}$.

Since, in the case of a purely data-driven policy $\hat{p}_{\boldsymbol{\theta}}$, the described control structure would lead to unpredictable robot behavior when $x$, $\phi$ or $c$ deviate from the trained distribution, we propose to formulate the context-adaptive policy as an MoE that combines an LfD policy with supplementary policies that keep the robot close to the demonstrated trajectories and enhance the convergence properties. While DS-based approaches that enforce constraints (e.g. stability [16]) on the LfD policy typically restrict inputs and outputs to the robot state and its derivatives, the combination of multiple policies enables the incorporation of additional, non-robot-related inputs into the LfD policy. Moreover, the modular structure of our MoE formulation allows policies to be added and removed ad-hoc, contributing to better interpretability.

As backbone of the context-adaptive policy framework (blue part in Fig. 1), we propose three ($P$) distinct policies, treated as Gaussian-distributed experts with means $\boldsymbol{\mu}_i$ and mixing coefficients $\pi_i$, forming an MoE with $\sum_{i=1}^{P} \pi_i = 1$:

- **LfD policy** ($\pi_{\mathrm{lfd}}$, $\boldsymbol{\mu}_{\mathrm{lfd}}$): A data-driven policy that is modulated in response to the current state of the robot and task-dependent parameters, which imitates the demonstrated motions.
- **Stabilizing policy** ($\pi_{\mathrm{sp}}$, $\boldsymbol{\mu}_{\mathrm{sp}}$): An uncertainty-aware policy that guides the robot back to regions where demonstrations are provided, overcoming the covariate shift and thus maintaining reliable performance.
- **Goal attractor policy** ($\pi_{\mathrm{gap}}$, $\boldsymbol{\mu}_{\mathrm{gap}}$): A kernel-based policy that ensures the robot motion converges to one of the demonstrated goal poses.

We define the control action $\hat{\boldsymbol{\xi}}_* = \begin{bmatrix} \dot{x}_d^\top & \dot{\phi}_d^\top \end{bmatrix}^\top$ at test input $s_*$ as the resulting MoE mean, computed by $\hat{\boldsymbol{\mu}} = \sum_{i=1}^{P} \pi_i \boldsymbol{\mu}_i$:

$$\hat{\boldsymbol{\xi}}_* = \hat{\boldsymbol{\mu}} = \pi_{\mathrm{lfd}} \boldsymbol{\mu}_{\mathrm{lfd}} + \pi_{\mathrm{sp}} \boldsymbol{\mu}_{\mathrm{sp}} + \pi_{\mathrm{gap}} \boldsymbol{\mu}_{\mathrm{gap}}. \quad (1)$$

In the following sections we describe the formulation of the means and mixing coefficients of the individual experts.

### A. LfD policy

In this work, we employ a multi-output GPR with independent output dimensions to generate the LfD policy, leveraging its data efficiency and explicit uncertainty formulation. Given $H$ demonstrated trajectories $\{\{s_{m,h}, \boldsymbol{\xi}_{m,h}\}_{m=1}^{M_h}\}_{h=1}^{H}$ with $M_h$ samples, we subsample $N$ data points $\{s_n, \boldsymbol{\xi}_n\}_{n=1}^{N}$, where the input $s_n = \begin{bmatrix} c_n^\top & x_n^\top & \phi_n^\top \end{bmatrix}^\top \in \mathbb{R}^{\mathcal{I}}$ encodes the end-effector pose and task-dependent parameters, and the output $\boldsymbol{\xi}_n = \begin{bmatrix} \dot{x}_n^\top & \dot{\phi}_n^\top \end{bmatrix}^\top \in \mathbb{R}^{\mathcal{O}}$ represents the corresponding linear and angular end-effector velocities. For a single test input $s_*$, the posterior predictive distribution yields a mean prediction $\boldsymbol{\mu}_* \in \mathbb{R}^{\mathcal{O}}$ and an associated predictive covariance $\Sigma_*$[2]:

$$\boldsymbol{\mu}_* = \boldsymbol{k}_*^\top (\boldsymbol{K} + \sigma^2 \boldsymbol{I}_N)^{-1} \boldsymbol{\Xi}, \quad (2)$$

$$\Sigma_* = k_{**} - \boldsymbol{k}_*^\top (\boldsymbol{K} + \sigma^2 \boldsymbol{I}_N)^{-1} \boldsymbol{k}_*. \quad (3)$$

Here, $\boldsymbol{\Xi}$ is defined as $\boldsymbol{\Xi} = \begin{bmatrix} \boldsymbol{\xi}_1 & \cdots & \boldsymbol{\xi}_N \end{bmatrix}^\top$, $\boldsymbol{I}_N$ is an identity matrix, $\boldsymbol{K}$ refers to the kernel matrix, $k_{**} = k(s_*, s_*)$ and $\boldsymbol{k}_* = \begin{bmatrix} k(s_*, s_1) & \cdots & k(s_*, s_N) \end{bmatrix}^\top$. In this work, the radial basis function (RBF) kernel, given by

$$k(s_i, s_j) = \exp\left( -\frac{1}{2}(s_i - s_j)^\top \boldsymbol{L}(s_i - s_j) \right), \quad (4)$$

is used. The length scale vector $\boldsymbol{l} = \begin{bmatrix} l_1 \cdots l_{\mathcal{I}} \end{bmatrix} \in \mathbb{R}^{\mathcal{I}}$, derived from $\boldsymbol{L} = \mathrm{diag}(\boldsymbol{l})^{-2}$, and the noise variance $\sigma^2$ denote the hyperparameters of GPR.

We define the predictive mean $\boldsymbol{\mu}_*$ as the LfD policy distribution mean, i.e. $\boldsymbol{\mu}_{\mathrm{lfd}} = \begin{bmatrix} \boldsymbol{\mu}_{\dot{x}}^\top & \boldsymbol{\mu}_{\dot{\phi}}^\top \end{bmatrix}^\top$. Its vector fields are exemplarily illustrated in Fig. 2a for the first letter of the LASA handwriting dataset [25], where we only consider $s = x \in \mathbb{R}^2$, i.e. no task-dependent parameter $c$.

---

[2]The covariance provides a closed-form epistemic uncertainty quantification, which increases with distance to the training data and therefore helps to identify out-of-distribution states, motivating the use of supplementary policies for improved stability.

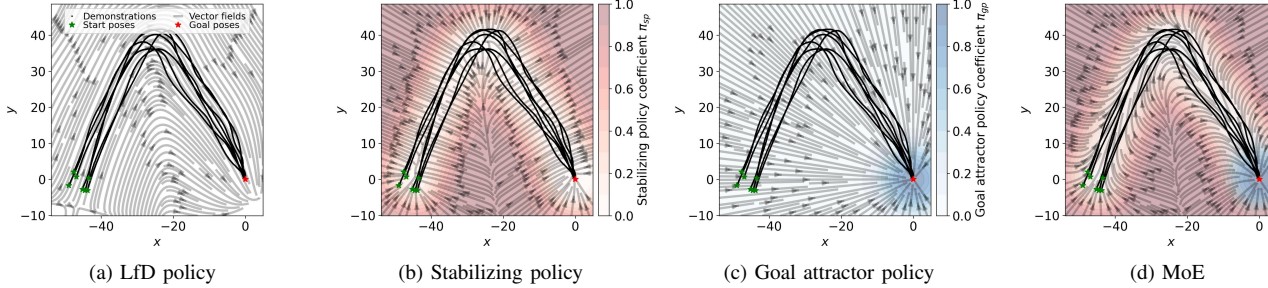

(a) LfD policy      (b) Stabilizing policy      (c) Goal attractor policy      (d) MoE

Fig. 2: Two-dimensional example showing the vector fields and the mixing coefficient activation of the MoE and the individual policies when trained on the first letter of the LASA handwriting dataset ('Angle'). (a) shows the LfD policy, (b) illustrates the stabilizing policy, (c) displays the goal attractor policy and (d) demonstrates the combination of the individual policies using the MoE formulation.

### B. Stabilizing policy

Due to the stability issues illustrated in Fig. 2a, the LfD policy is only guaranteed to reproduce the demonstrated behavior along the learned trajectories. Therefore, we introduce a stabilizing policy that automatically guides the robot back to the demonstrations in case of deviations or different initial conditions. Based on the covariance prediction of GPR (Eq. (3)), which provides a closed-form epistemic uncertainty quantification, that increases smoothly with the distance to the training data, we formulate a policy, similar to [21], [22] but with orientation incorporated, that generates actions in the direction of gradient descent. For this, we derive the gradient of $\Sigma_*$ with respect to the current state $s_*$[3], assuming the RBF kernel Eq. (4):

$$\nabla\Sigma_* = -2\nabla\boldsymbol{k}_*^\top(\boldsymbol{K}+\sigma^2\boldsymbol{I}_N)^{-1}\boldsymbol{k}_*, \qquad (5)$$

where $\nabla\boldsymbol{k}_* = \left[\nabla k(\boldsymbol{s}_*,\boldsymbol{s}_1)^\top \cdots \nabla k(\boldsymbol{s}_*,\boldsymbol{s}_N)^\top\right]^\top$ and $\nabla k(\boldsymbol{s}_*,\boldsymbol{s}_n) = -k(\boldsymbol{s}_*,\boldsymbol{s}_n)\boldsymbol{L}(\boldsymbol{s}_*-\boldsymbol{s}_n)$ is the gradient of the RBF kernel. In this work we assume that, in contrast to the robot state, the task-dependent parameter $\boldsymbol{c}$ can only be controlled indirectly and thus focus on the robot state dependent terms $\nabla_{\boldsymbol{x}_*}\Sigma_*, \nabla_{\boldsymbol{\phi}_*}\Sigma_*$ to design the stabilizing policy[4]. To regulate the magnitude of the velocities, we normalize the gradients for position and orientation independently and multiply them by the respective covariance $\Sigma_{\boldsymbol{c},\boldsymbol{x}}, \Sigma_{\boldsymbol{c},\boldsymbol{\phi}}$, which are computed using Eq. (3) with $\boldsymbol{s} = \left[\boldsymbol{c}^\top \boldsymbol{x}^\top\right]^\top$ and $\boldsymbol{s} = \left[\boldsymbol{c}^\top \boldsymbol{\phi}^\top\right]^\top$ respectively. This leads to a vanishing velocity towards the associated demonstrated trajectories. Further, we introduce the hyperparameter $\boldsymbol{K}_{\mathrm{sp}} = \{K_{\mathrm{sp,x}}, K_{\mathrm{sp},\boldsymbol{\phi}}\}$ that maps the normalized velocities to the domain and define the mean of the stabilizing policy distribution as:

$$\boldsymbol{\mu}_{\mathrm{sp}} = \begin{bmatrix} -K_{\mathrm{sp,x}}\frac{\nabla_{\boldsymbol{x}_*}\Sigma_*}{\|\nabla_{\boldsymbol{x}_*}\Sigma_*\|}\Sigma_{\boldsymbol{c},\boldsymbol{x}} \\ -K_{\mathrm{sp},\boldsymbol{\phi}}\frac{\nabla_{\boldsymbol{\phi}_*}\Sigma_*}{\|\nabla_{\boldsymbol{\phi}_*}\Sigma_*\|}\Sigma_{\boldsymbol{c},\boldsymbol{\phi}} \end{bmatrix}(\Delta t)^{-1}, \qquad (6)$$

where $\Delta_t^{-1}$ is the outer loop control rate. Equation (6) guides the robot towards regions with low epistemic uncertainty. This effect can be observed on the vector fields in Fig. 2b.

---

[3]For simplicity and improved readability, we write $\nabla$ instead of $\nabla_{\boldsymbol{s}_*}$.
[4]Note that for an input of the form of $\boldsymbol{s}$, the gradient contains three distinct terms, $\nabla\Sigma_* = \left[(\nabla_{\boldsymbol{c}_*}\Sigma_*)^\top (\nabla_{\boldsymbol{x}_*}\Sigma_*)^\top (\nabla_{\boldsymbol{\phi}_*}\Sigma_*)^\top\right]^\top$.

### C. Goal attractor policy

Since GPR is not guaranteed to converge to the demonstrated goal poses $\boldsymbol{s}_{M_h,h}$, as can be seen in Fig. 2a, we further introduce a policy that guarantees this convergence. Defining $\boldsymbol{s}_g = \arg\max\{k(\boldsymbol{s}_*,\boldsymbol{s}_{M_1,1}),\ldots,k(\boldsymbol{s}_*,\boldsymbol{s}_{M_H,H})\}$ as the demonstrated goal pose with the highest kernel activation at current state $\boldsymbol{s}_*$, we design the mean of this policy using the gradient of the corresponding kernel, i.e. $\nabla k(\boldsymbol{s}_*,\boldsymbol{s}_g)$ with $\boldsymbol{s}_g = \left[\boldsymbol{c}_g^\top \boldsymbol{x}_g^\top \boldsymbol{\phi}_g^\top\right]^\top$. As with the stabilizing policy, we normalize the gradients for position and orientation independently and introduce the hyperparameter $\boldsymbol{K}_{\mathrm{gap}} = \{K_{\mathrm{gap,x}}, K_{\mathrm{gap},\boldsymbol{\phi}}\}$ to map the resulting velocities to the domain. In contrast, we now use $k(\boldsymbol{x}_*,\boldsymbol{x}_g)$ and $k(\boldsymbol{\phi}_*,\boldsymbol{\phi}_g)$ to ensure vanishing velocities towards the goal pose. We define the mean of the goal attractor policy distribution as:

$$\boldsymbol{\mu}_{\mathrm{gap}} = \begin{bmatrix} K_{\mathrm{gap,x}}\frac{\nabla_{\boldsymbol{x}_*}k(\boldsymbol{s}_*,\boldsymbol{s}_g)}{\|\nabla_{\boldsymbol{x}_*}k(\boldsymbol{s}_*,\boldsymbol{s}_g)\|}(1-k(\boldsymbol{x}_*,\boldsymbol{x}_g)) \\ K_{\mathrm{gap},\boldsymbol{\phi}}\frac{\nabla_{\boldsymbol{\phi}_*}k(\boldsymbol{s}_*,\boldsymbol{s}_g)}{\|\nabla_{\boldsymbol{\phi}_*}k(\boldsymbol{s}_*,\boldsymbol{s}_g)\|}(1-k(\boldsymbol{\phi}_*,\boldsymbol{\phi}_g)) \end{bmatrix}(\Delta t)^{-1}. \quad (7)$$

The vector fields resulting from Eq. (7) are visualized in Fig. 2c for the considered illustrative example.

### D. MoE mixing coefficient design

We here outline the design choices for the mixing coefficients of Eq. (1) that are automatically computed based on the uncertainty inherent in the provided demonstrations.

*1) Goal attractor policy coefficient $\pi_{\mathrm{gap}}$:* Close to the respective goal, we want to ensure convergence and, hence, an activation that smoothly increases as the distance to the goal pose decreases. Therefore, we define

$$\pi_{\mathrm{gap}} = k(\boldsymbol{s}_*,\boldsymbol{s}_g) \qquad (8)$$

as the goal attractor policy coefficient, where $\boldsymbol{s}_g$ is recomputed at each iteration based on the current state $\boldsymbol{s}_*$.

*2) Stabilizing policy coefficient $\pi_{\mathrm{sp}}$:* Furthermore, we assign a high priority to staying close to demonstrations, avoiding dangerous behaviors. Therefore, we use the covariance prediction provided by GPR (Eq. (3)), which increases smoothly when deviating from the data, as activation.

We reduce the mixing coefficient by the activation of the goal attractor policy, as we assign a higher priority to convergence close to the goal. We thus define

$$\pi_{\mathrm{sp}} = (1-\pi_{\mathrm{gap}})\Sigma_* = (1-k(\boldsymbol{s}_*,\boldsymbol{s}_g))\Sigma_*. \qquad (9)$$

*3) LfD policy coefficient $\pi_{\text{lfd}}$:* Taking into account the normalization $\sum_{i=1}^{P} \pi_i = 1$ results in

$$\pi_{\text{lfd}} = 1 - \pi_{\text{gap}} - \pi_{\text{sp}} = (1 - k(\boldsymbol{s}_*, \boldsymbol{s}_g))(1 - \Sigma_*) \quad (10)$$

as the mixing coefficient for the LfD policy. Hence, the LfD policy is smoothly activated near the demonstrations, while the goal attractor policy takes over close to the goal.

The mixing coefficients for each policy as well as the combined MoE output $\hat{\boldsymbol{\mu}}$, applied to the 2D example, are displayed in Fig. 2 using contour lines. Here, each policy acts as an expert in its respective area, while the combined policy unifies the advantages of the individual policies.

Algorithm I.1 summarizes this section in pseudo-code.

## III. EVALUATION

We evaluate our approach both in simulation and three real robot experiments. The hyperparameter values used are obtained through optimization as described in Appendix II.

### A. LASA handwriting dataset

First, we investigate our approach's performance on the LASA handwriting dataset [25] and compare it with different combinations of the individual policies and two established baseline approaches: SEDS [16] and Diffusion Policy [14]. In this experiment, we focus on the reproduction of the demonstrated handwriting motions with different initial conditions, showing the approach's stability and convergence properties. Thus, we define $\boldsymbol{s} = \boldsymbol{x} \in \mathbb{R}^2$, $\boldsymbol{\xi} = \dot{\boldsymbol{x}} \in \mathbb{R}^2$, omitting the task-dependent parameter $\boldsymbol{c}$. We perform two sets of simulations, similar to those described in Appendix II, for all letters: one in which $\boldsymbol{x}_{0,k}$ is placed at the initial pose of each of the seven demonstrations and one in which $\boldsymbol{x}_{0,k}$ is placed ten times randomly within the input space. Table I shows the averaged results over five seeds.

TABLE I: Quantitative evaluation (average success $S$, iterations $I$, distance $D$) on the LASA handwriting dataset (lower is better for $I$ and $D$)

| Exp. | Method | $S$ [%] | $I$ | $D$ |
|---|---|---|---|---|
| $\boldsymbol{x}_{0,k}$ on demos | LfD (II-A) | 7.0 | 471.5 | 5.00 |
| | LfD & stab. (II-B) | 63.8 | 264.9 | 0.90 |
| | LfD & goal attr. (II-C) | 99.8 | 133.0 | 0.63 |
| | **Proposed framework** | **100.0** | 151.9 | **0.47** |
| | Diffusion Policy ([14]) | 8.0 | 482.7 | 42.79 |
| | SEDS (likelihood, [16]) | 99.6 | **126.6** | 3.63 |
| $\boldsymbol{x}_{0,k}$ random | LfD (II-A) | 3.5 | 486.1 | 9.81 |
| | LfD & stab. (II-B) | 63.1 | 236.2 | 1.62 |
| | LfD & goal attr. (II-C) | 76.5 | 220.5 | 5.47 |
| | **Proposed framework** | **99.7** | 116.6 | **1.03** |
| | Diffusion Policy ([14]) | 8.8 | 476.5 | 60.70 |
| | SEDS (likelihood, [16]) | 99.4 | **100.6** | 3.51 |

### B. Real robot experiments

We then highlight our approach's adaptability, robustness, and real-time reactivity in three real robot manipulation scenarios using a 7-DoF KUKA LWR robotic arm[5]:

---
[5]Detailed information about the experiments can be found in Appendix III. A video showing all experiments can be accessed at this link.

*1) Force-conditioned grasping:* The robot picks up and transports a 3D-printed sphere while dynamically adjusting its behavior based on tactile sensor readings in the fingertips, i.e. $\boldsymbol{c} = [f_{\text{left}} \ f_{\text{right}}]^\top$. The learned policy not only reproduces the demonstrated trajectories but also re-grasps the object if dropped, showcasing its reactivity to unexpected changes. The quantitative results presented in Table II highlight superior success rates ($S$), iteration efficiency ($I$), and collision avoidance (*Coll.*) compared to baseline methods.

*2) Manipulation of deformable food items:* The robot places a fish fillet on a tray, adapting to its in-gripper geometry, which is represented by the fillet's overlap on each side of the gripper, i.e. $\boldsymbol{c} = [d_{\text{left}} \ d_{\text{right}}]^\top$. The learned policy performs placement differently depending on the grasp configuration, ensuring accurate placement (see Table II).

*3) Dynamic grasping:* The robot grasps YCBV objects from a conveyor belt, adapting its behavior based on task phases and the object to be grasped, i.e. $\boldsymbol{c} = [k_{\text{obj}} \ j_{\text{phase}}]^\top$. The learned policy is able to handle various dynamic grasping scenarios, including mid-execution object swapping and human-robot handovers, while maintaining robustness.

TABLE II: Quantitative evaluation (average success $S$, iterations $I$, distance $D$, collisions *Coll.*) on the real robot experiments in Section III-B

| Exp. | Method | $S$ [%] | $I$ | $D$ | *Coll.* |
|---|---|---|---|---|---|
| Sec. III-B.1 | LfD (II-A) | 0.0 | 1000.0 | 0.096 | 16 |
| | LfD & stab. (II-B) | 55.0 | 694.2 | 0.022 | **0** |
| | LfD & goal attr. (II-C) | 5.0 | 851.5 | 0.088 | 16 |
| | **Proposed framework** | **100.0** | **264.3** | **0.016** | **0** |
| | Diffusion Policy ([14]) | 0.0 | – | 0.076 | 20 |
| Sec. III-B.2 | LfD (II-A) | 0.0 | 1000.0 | 0.148 | 7 |
| | LfD & stab. (II-B) | 0.0 | 1000.0 | 0.024 | **0** |
| | LfD & goal attr. (II-C) | 20.0 | 839.6 | 0.097 | 3 |
| | **Proposed framework** | **95.0** | **370.1** | **0.019** | **0** |
| | Diffusion Policy ([14]) | 0.0 | 1000.0 | 0.211 | 19 |

## IV. DISCUSSION AND CONCLUSION

This work shows that even established policy learning methods, such as GPR and DP, often fail to exhibit robust behavior, particularly in small-data regimes, leading to high collision rates and frequent timeouts (see Table II).

By contrast, our context-adaptive policy framework (Section II) achieves high success and low collision rates while maintaining proximity to the demonstrated data (see Tables I and II). Unlike SEDS, our approach allows for dynamic adaptation to continuous (e.g., contact forces) and discrete (e.g., task phases) task-dependent parameters, enabling context-adaptive manipulation. The conducted experiments on a 7-DoF robot, validate its data efficiency and robustness.

Limitations include the task-specific parameter representation, scalability constraints due to kernel-based methods, and GPR's input dimensionality restrictions. However, the modularity of our formulation permits the incorporation of alternative LfD policy representations. In future work, we will investigate the use of such representations that can handle more diverse inputs (e.g., diffusion policies or flow-based policies) while leveraging the robustness of our framework.

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

## APPENDIX I
### ALGORITHMIC IMPLEMENTATION OF OUR APPROACH

Algorithm I.1 formalizes the learning and execution procedure using our framework, as introduced in Section II.

---

**Algorithm I.1** Skill learning and execution

---

**Input:** $\{\{\boldsymbol{s}_{m,h}, \boldsymbol{\xi}_{m,h}\}_{m=1}^{M_h}\}_{h=1}^{H}$, $\{\boldsymbol{l}, \sigma^2, \boldsymbol{K}_{\text{sp}}, \boldsymbol{K}_{\text{gap}}\}$, $\boldsymbol{s}_0$

$\{\boldsymbol{s}_n, \boldsymbol{\xi}_n\}_{n=1}^{N} \subset \{\{\boldsymbol{s}_{m,h}, \boldsymbol{\xi}_{m,h}\}_{m=1}^{M_h}\}_{h=1}^{H}$ ▷ Subsample

$\boldsymbol{R} \leftarrow \text{chol}(\boldsymbol{K} + \sigma_n^2 I)$

$\boldsymbol{\alpha} \leftarrow \boldsymbol{R}^\top \backslash (\boldsymbol{R} \backslash \boldsymbol{\Xi})$ ▷ Train GP (cf. [24], Alg. 2.1)

$\boldsymbol{s}_* \leftarrow \boldsymbol{s}_0$ ▷ Assign initial conditions

$i \leftarrow 0$ ▷ Reset iteration counter

converged ← false

stopTriggered ← false

**while** ¬ converged $\wedge$ $i < I_{\max}$ $\wedge$ ¬ stopTriggered **do**

    $\boldsymbol{\mu}_{\text{lfd}} \leftarrow$ Eq. (2) ▷ Compute GP mean

    $\boldsymbol{\mu}_{\text{sp}} \leftarrow$ Eq. (6) ▷ Calculate stab. policy mean

    $\boldsymbol{\mu}_{\text{gap}} \leftarrow$ Eq. (7) ▷ Determine goal attr. policy mean

    $\pi_{\text{gap}} \leftarrow$ Eq. (8) ▷ Calculate goal attr. mixing coeff.

    $\pi_{\text{sp}} \leftarrow$ Eq. (9) ▷ Determine stab. mixing coeff.

    $\pi_{\text{lfd}} \leftarrow$ Eq. (10) ▷ Derive LfD mixing coeff.

    $\hat{\boldsymbol{\xi}}_* \leftarrow$ Eq. (1) ▷ Compute control action (MoE mean)

    $\boldsymbol{s}_* \leftarrow$ InnerControlLoop($\hat{\boldsymbol{\xi}}_*$) ▷ Apply $\hat{\boldsymbol{\xi}}_*$ to robot

    converged ← checkConvergence($\boldsymbol{s}_*$)

    stopTriggered ← checkExternalStop()

    $i \leftarrow i + 1$ ▷ Increment iteration counter

**end while**

---

## APPENDIX II
### HYPERPARAMETER OPTIMIZATION

Given the influence of GPR hyperparameters on the supplementary policies (e.g. through the shared kernel length over all policies), the optimization of the combined hyperparameters $\boldsymbol{\theta} = \{\boldsymbol{l}, \sigma^2, \boldsymbol{K}_{\text{sp}}, \boldsymbol{K}_{\text{gap}}\}$ is performed jointly using the CMA-ES algorithm [26], which is used due to its derivative-free nature and its ability to efficiently search complex parameter spaces. To this end, the skill learning and execution procedure, as summarized in Algorithm I.1, is executed ten times for the provided demonstrations, while

TABLE I.1: Experiment-dependent parameters and hyperparameters used throughout experiments. Vector-valued hyperparameters are split into individual columns. Values apply to both the full framework and combinations of individual policies.

| Exp. | $\mathcal{I}$ | $\mathcal{O}$ | $N$ | $\Delta_t^{-1}$ | $I_{\max}$ | Seed | $\sigma^2$ | $l_c$ | $l_x$ | $l_\phi$ | $K_{\text{sp,x}}$ | $K_{\text{sp},\phi}$ | $K_{\text{gap,x}}$ | $K_{\text{gap},\phi}$ |
|---|---|---|---|---|---|---|---|---|---|---|---|---|---|---|
| Sec. III-A | 2 | 2 | 500 | 20 | 500 | 0 | 1.720 | - | 4.039 | - | 63.054 | - | 119.136 | - |
| | | | | | | 1 | 2.443 | - | 3.371 | - | 42.112 | - | 27.001 | - |
| | | | | | | 2 | 42.602 | - | 11.340 | - | 1.699 | - | 101.386 | - |
| | | | | | | 3 | 3.031 | - | 6.582 | - | 68.922 | - | 89.673 | - |
| | | | | | | 4 | 52.539 | - | 9.892 | - | 9.458 | - | 117.239 | - |
| Sec. III-B.1 | 8 | 7 | 500 | 20 | 1000 | 42 | 0.432 | 8.095 | 0.066 | 0.074 | 0.280 | 0.627 | 0.281 | 0.782 |
| Sec. III-B.2 | 8 | 6 | 500 | 20 | 1000 | 42 | 0.432 | 20.095 | 0.066 | 0.074 | 0.280 | 0.627 | 0.281 | 0.782 |
| Sec. III-B.3 | 8 | 7 | 500 | 20 | – | 42 | 0.432 | 0.095 | 0.066 | 0.074 | 0.280 | 0.627 | 0.281 | 0.782 |

the robot behavior (*InnerControlLoop*) is simulated by computing the next pose at each time step using $\boldsymbol{x}_{t+1,i} = \boldsymbol{x}_{t,i} + \Delta t \dot{\boldsymbol{x}}_{t,i}$ and $\boldsymbol{\phi}_{t+1,i} = \boldsymbol{\phi}_{t,i} + \Delta t \dot{\boldsymbol{\phi}}_{t,i}$. To optimize both the behavior along demonstrated trajectories and system stabilization, the starting poses $\{\boldsymbol{x}_{0,k}, \boldsymbol{\phi}_{0,k}\}_{k=1}^{10}$ are sampled randomly within the input space.

Each of the ten simulated trials is terminated when the robot has converged or $I_{\max} = 500$ iterations are reached. Defining $\boldsymbol{p} = \begin{bmatrix} \boldsymbol{x}^\top & \boldsymbol{\phi}^\top \end{bmatrix}^\top \in \mathbb{R}^\mathcal{P}$ as the end-effector pose, convergence is achieved if each component $p_{*,r}$ of the current end-effector pose $\boldsymbol{p}_*$ remains for ten iterations within the distance $d_r = 0.01 \cdot d_{\max,r}$ to one of the demonstrated goal poses $\{\boldsymbol{p}_{M_h,h}\}_{h=1}^H$, with $d_{\max,r} = \max\limits_{m,h} p_{m,h,r} - \min\limits_{m,h} p_{m,h,r}$ describing the max-min range across all demonstrations for the specific entry and $r \in [0, \mathcal{P}]$. We determine converged trials as successful $S_k = 1$ and non-converged trials as unsuccessful $S_k = 0$. Further, we measure the number of iterations $i_k \in [0, I_{\max}]$ and the mean distance between the executed poses and the closest demonstrated poses across all time steps $D_k = \frac{1}{i_k} \sum_{j=0}^{i_k} \min\limits_{m,h} \|\boldsymbol{p}_{m,h} - \boldsymbol{p}_j\|$. We define the multi-objective cost function as an equally weighted sum:

$$\min_{\boldsymbol{\theta}} J = (1 - S) + \frac{I}{I_{\max}} + \frac{D}{\sqrt{\sum_{r=1}^{\mathcal{P}} d_{\max,r}^2}}, \qquad \text{(II.1)}$$

where, $S$, $I$ and $D$ are the average success, iterations and distance over ten trials, thus all objectives lie within $[0, 1]$. We further constrain the search space of the hyperparameters in $\boldsymbol{\theta}$ to $1\% - 100\%$ of their respective max-min ranges.

The hyperparameters for the LASA experiments (Section III-A) are obtained using 100 optimization iterations on five random letters. As the robot state and action spaces remain the same for the real robot experiments (Section III-B), we optimize the hyperparameters over 500 optimization iterations using the training data of Section III-B.3 and then manually adjust the task-parameter-dependent kernel length for Sections III-B.1 and III-B.2 as only the domain of $\boldsymbol{c}$ changes. Table I.1 summarizes all experiment-dependent parameters and hyperparameters used throughout the experiments described in Section III-B, for both the full framework and combinations of individual policies. For the LASA experiments (Section III-A), five random seeds were used, whereas for the real robot experiments a single seed was used. Unused hyperparameters are indicated by "–".

TABLE II.2: Diffusion Policy hyperparameters used across all experiments.

| Hyperparameter | Value |
|---|---|
| Observation horizon, $T_o$ | 2 |
| Prediction horizon, $T_p$ | 16 |
| Action horizon, $T_a$ | 8 |
| Diffusion steps (train), D-Iters Train | 100 |
| Diffusion steps (eval), D-Iters Eval | 10 |
| Number of training epochs | 100 |
| Total number of NN parameters | $\sim 65.3$ M |

Table II.2 lists the hyperparameters used for Diffusion Policy (*state-based environment*[6]) across all experiments. The notation for the hyperparameters is adopted from [14].

## APPENDIX III
### DETAILS ABOUT THE REAL ROBOT EXPERIMENTS

This appendix provides a more detailed account of the experimental setup, procedures, and additional results that complement the summary presented in the main text.

### A. Force-conditioned grasping

In the second experiment, we demonstrate our approach's robust transferability to real robots and its ability to react to changing task-parameter values at runtime. For this, we record ten trajectories via kinesthetic teaching using a 7-DoF KUKA LWR robotic arm equipped with a variable stiffness parallel gripper, where the robot picks up a 3D-printed sphere from a table and transports it to a demonstrated goal pose (Fig. III.1a). We make use of the incorporated tactile sensors in both fingertips to derive the forces exerted by the object and define $\boldsymbol{c} = [f_{\text{left}}\ f_{\text{right}}]^\top$ as task-parameter. In addition, the input includes the robot position and orientation as Euler vector, i.e. $\mathcal{I} = 8$. The demonstrated and reproduced task-parameters can be seen in Fig. III.1b.

Similar to the experiments in Section III-A, we quantitatively evaluate our approach and different combinations of the individual policies on this task and compare it with Diffusion Policy [14][7]. For this, we perform 20 trials, where the initial poses $\{\boldsymbol{x}_{0,k}, \boldsymbol{\phi}_{0,k}\}$ are randomly placed within the robot task space, as shown in Fig. III.1e. Our context-adaptive policy, whose vector fields are exemplarily displayed for a single trial at $\boldsymbol{s}_0$ (Fig. III.1c) and $\boldsymbol{s}_I$ (Fig. III.1d),

---

[6]Available at https://github.com/real-stanford/diffusion_policy.

[7]A comparison with SEDS, as in Section III-A, is not feasible in this and the subsequent experiments, since SEDS is restricted to specific input and output variables due to the incorporation of hard stability constraints.

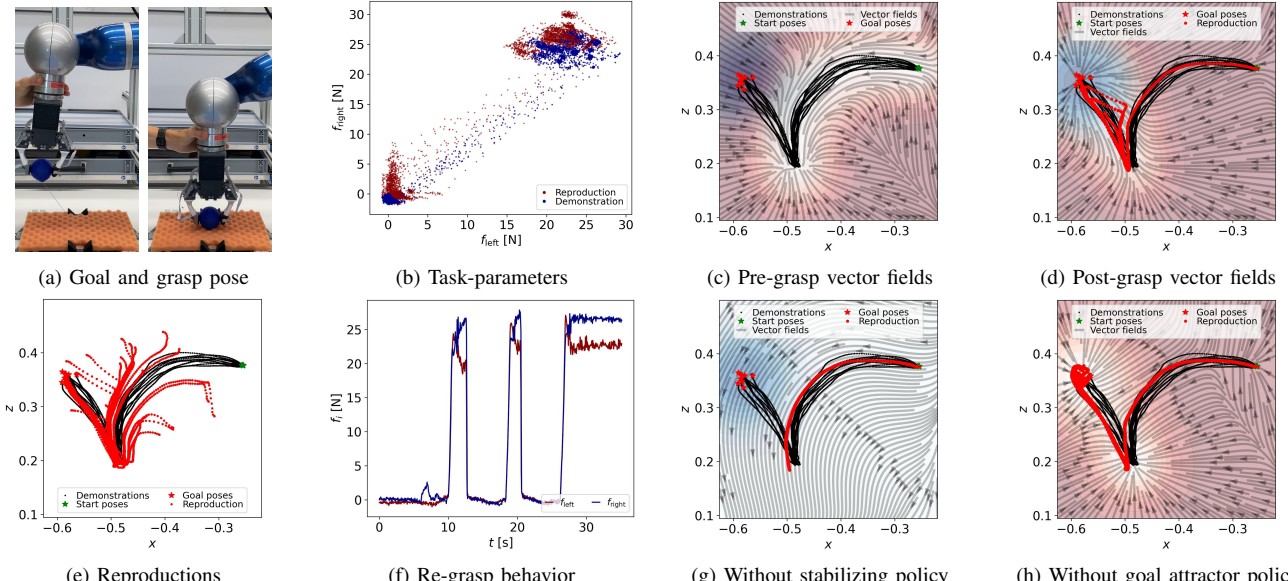

| | | | |
|---|---|---|---|
| (a) Goal and grasp pose | (b) Task-parameters | (c) Pre-grasp vector fields | (d) Post-grasp vector fields |
| (e) Reproductions | (f) Re-grasp behavior | (g) Without stabilizing policy | (h) Without goal attractor policy |

Fig. III.1: Demonstrations and reproductions of the force-conditioned grasping experiment: (a) illustrates the demonstrated grasp and goal pose, (b) shows the reproduced trajectories from the quantitative evaluation (upper part of Table II) in the x-z-plane, (c) illustrates the demonstrated and reproduced task-parameters from the quantitative evaluation, (d) presents the reproduced task-parameter evolution for a trial with multiple re-grasps where the gripper is manually opened by the user, (e), (f) display the pre- and post-grasp MoE vector fields at $\boldsymbol{s}_0$ and $\boldsymbol{s}_I$ for this trial and (g), (h) display the vector fields and observed robot trajectories when removing either the stabilizing or the goal attractor policy within the context-adaptive policy framework.

is used to compute the end-effector velocity $\dot{\boldsymbol{x}}_d$, $\dot{\boldsymbol{\phi}}_d$ and the gripper joint velocity $\dot{q}_g$ in a 20 Hz loop, hence $\mathcal{O} = 7$. Due to the longer task horizon compared to Section III-A, we define $I_{\max} = 1000$, and add a 1 mm or 1° offset to $d_r$, depending on whether it represents position or orientation, to account for measurement inaccuracies. Since this is a real-world task, in addition to $S$, $I$ and $D$, we use the number of collisions with the environment (*Coll.*) as comparison metric. The results are displayed in Table II and Fig. III.1e.

Furthermore, we demonstrate the approach's reactivity by opening the gripper manually such that the object drops from the gripper. Due to the task-parameter conditioning, the robot attempts to re-grasp the sphere at its original place where it is pulled back via a cable. The resulting task-parameters are exemplified in Fig. III.1f for the case where the gripper is opened twice. The corresponding trajectory and the vector fields are displayed in Fig. III.1d. Moreover, we show that the proposed combination of expert policies is essential for accurate reproduction by removing either the stabilizing (Fig. III.1g) or the goal attractor policy (Fig. III.1h).

### B. Manipulation of deformable food items

In the third experiment, we demonstrate our approach's adaptation to different placing strategies of a deformable object, in this case a fish fillet. Using a similar robot and gripper as in Section III-B.1, but with different fingertips in order to facilitate the grasp, we record the placing of the fish on a tray via kinesthetic teaching. Depending on the grasp configuration, the placing is performed differently, by approaching the tray from the side where the fillet hangs. We provide four demonstrations for each side. To capture the in-gripper fish geometry, we fine-tune YOLOv7 [27] on the described use-case to obtain two bounding boxes, one

for each side of the gripper. We define the task-parameters as the fish's overlap on each side of the gripper, i.e. $\boldsymbol{c} = [d_{\text{left}}\ d_{\text{right}}]^{\top}$, where $d_i = \max_k \left\| [u_{k,i}\ v_{k,i}]^{\top} - [u_{ee}\ v_{ee}]^{\top} \right\|$ is the maximum distance between the end-effector pose $[u_{ee}\ v_{ee}]^{\top}$, transformed to image coordinates, and the $k \in [0, 4]$ corners of the respective bounding box $[u_{k,i}\ v_{k,i}]^{\top}$[8]. Fig. III.2a clarifies the definition of $\boldsymbol{c}$ and Fig. III.2c shows the demonstrated task-parameters together with the reproduced ones. Including robot position and orientation, the input dimension is $\mathcal{I} = 8$.

We perform an identical evaluation as in Section III-B.1, where the context-adaptive policy is used to compute the end-effector velocity $\{\dot{\boldsymbol{x}}_d, \dot{\boldsymbol{\phi}}_d\}$ in a 20 Hz loop, hence $\mathcal{O} = 6$. The results are shown in Table II and the reproduced task-parameters are illustrated in Fig. III.2c. Moreover, Fig. III.2e displays reproduced trajectories for both fish fillet configurations when starting at the demonstrated start poses and Fig. III.2b, Fig. III.2d and Fig. III.2f exemplify the vector fields and the reproduced trajectory of a single trial at different times in case the fish fillet hangs on the right.

### C. Dynamic grasping

In the final experiment, we apply our approach to dynamically grasp four selected YCBV objects[9] from a conveyor belt. We record three demonstrations per object via kinesthetic teaching with static conveyor belt consisting of a *grasping*, *placing* and *resting* phase. During the *grasping*

---

[8]As this work focuses on the context-adaptive policy framework, the deformable object detection is implemented in this simplified, but sufficient form. However, more complex or 3D-based detectors would also be conceivable if the task-parameter dimension remains treatable by GPR.

[9]Available at https://rse-lab.cs.washington.edu/projects/posecnn/.

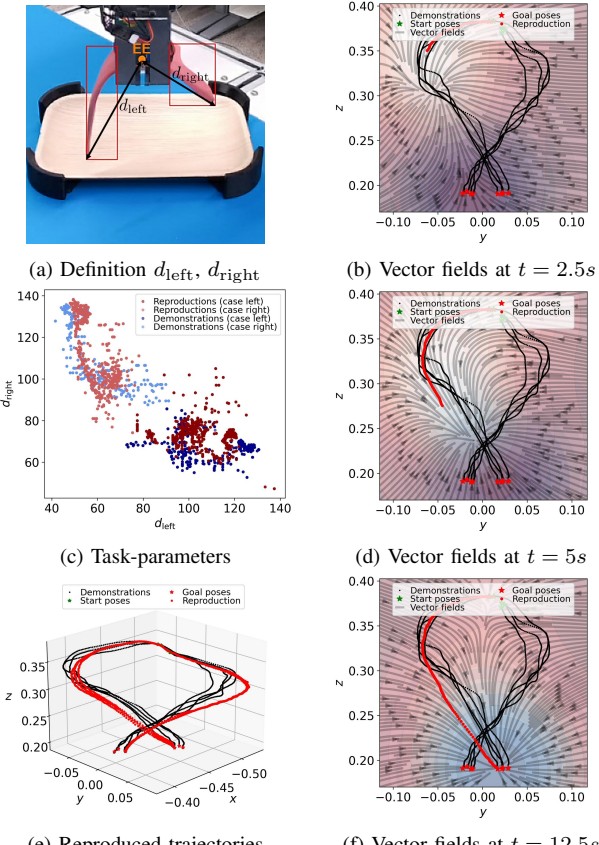

(a) Definition $d_{\text{left}}$, $d_{\text{right}}$

(b) Vector fields at $t = 2.5s$

(c) Task-parameters

(d) Vector fields at $t = 5s$

(e) Reproduced trajectories

(f) Vector fields at $t = 12.5s$

Fig. III.2: Demonstrations and reproductions of the manipulation of deformable food items experiment: (a) clarifies the definition of the task-parameters, (c) displays the reproduced trajectories of each four trials for both fish fillet configurations when starting at the demonstrated start poses, (e) shows the demonstrated and reproduced task-parameters from the quantitative evaluation (lower part of Table II) and (b), (d), (f) exemplify the vector fields and the reproduced trajectory of a single trial at $t = 2.5s$, $t = 5s$ and $t = 12.5s$ respectively when the fish fillet hangs on the right.

another on the remaining two datasets in world coordinate system. Depending on the current phase, the respective policy is selected, allowing the object to be grasped at any location within the workspace while the *placing* and *resting* poses remain static. Moreover, a non-zero mean is used for the gripper joint velocity within the *grasping* phase so that the gripper re-opens while approaching the object. Finally, a Matérn kernel [24] is used for $k(\boldsymbol{x}_*, \boldsymbol{x}_g)$ and $k(\boldsymbol{\phi}_*, \boldsymbol{\phi}_g)$ in Eq. (7), producing higher end-effector velocities near the goal poses and thus allowing higher object velocities.

We demonstrate the robot's capability to dynamically grasp all four selected objects at various poses on both static and moving conveyor belts, with varying initial positions of both the objects and the robot. Furthermore, the experiment highlights the robot's adaptive behavior during mid-execution object swapping and human-robot handover scenarios, where the robot continuously tracks and aligns with the object's changing position in the human hand to successfully complete the grasp. The approach also demonstrates robustness against external disturbances. More detailed visualizations of these scenarios are provided in the accompanying video.

phase the object is approached and grasped within a pre-defined region of the workspace, before being placed again on the conveyor belt outside the pre-defined region during the *placing* phase. After placing the object, the robot moves to a defined position in which it waits (*resting* phase) until the object moves back into the pre-defined region.

The object position is tracked at 20Hz using a combination of learning and model-based computer vision methods, including YOLOv7 [27], a 6D object pose estimation [28] and a 3D Object Tracking [29]. We define the task-parameters as $\boldsymbol{c} = [k_{\text{obj}} \, j_{\text{phase}}]^\top$, where $k_{\text{obj}} \in \{2, 5, 7, 9\}$ represents "Cracker Box", "Mustard Bottle", "Pudding Box" and "Potted Meat Can". Further, $j_{\text{phase}} \in \{0, 1, 2\}$ represents the *grasping*, *placing* and *resting* phase, that are active when the object is located within the pre-defined region, grasped or located outside the pre-defined region, respectively. Thus, the input and output dimensions are $\mathcal{I} = 8$ and $\mathcal{O} = 7$.

To robustly accomplish the task, three adaptations were made compared to previous experiments. First, the continuous demonstrations were divided by phase into three datasets. Two separate policies were trained: one on the *grasping* phase dataset transformed to object coordinate system, and