# OpenReview forum: "Context-adaptive policy framework for robust and reactive robotic manipulation via uncertainty-aware imitation learning"
_IEEE.org/ICRA/2026/Workshop/Manipulation_Robustness — ICRA 2026_

### Official Review · Reviewer_Ryjc · 2026-05-04
**Interesting work and valuable research question. More discussion on limitations required and room for improvement in experiment presentation.**

**Rating:** 7
**Confidence:** 3

**Review:**

This paper tries to improve the generalization of imitation learning policies under low data regimes, which is a very valuable research direction. Although I have some questions about the applicability of the framework for vision-based tasks, I believe this paper would be an interesting contribution to the workshop.

The authors combine a Gaussian-Process policy with two heuristic policies in a MoE-style composition. From their ablations, both improve performance. I think the authors should elaborate more on the limitations of the current framework and how it could be used to tackle tasks for which we don't have a low-dimensional state representation, in which case GPR will not suffice. How can we then still obtain a calibrated and accurate uncertainty estimation for the stabilizing policy and a similarity metric for the goal attractor? As far as I know, both are not trivial with generative action heads such as diffusion or flow matching. Some interesting references might be the use of time-to-completion prediction for goal proximity and the use of the diffusion training objective as a measure for uncertainty (https://arxiv.org/abs/2410.14868).

The experiments are interesting, but I think they could be presented in a more informative way. In particular, I think some images/illustrations would help to show:
	- what is the exact task?
	- what are the initial conditions of the demonstrations? (potentially showing an alpha composition of the initial scenes)
This will both help to assess the difficulty of the tasks and performance differences. The video was very helpful in this regard, but I think this is important and should hence be in the main text.

Finally, I would expect the Diffusion policy baseline to perform better. Perhaps the authors could provide more hyperparameter details and share insight as to why DP does not work here. For example, does performance improve with more demonstrations? And if so, how do both methods scale with the number of demonstrations? This would validate the assumption that it is because of the extremely low number of demonstrations. If not, then please show how you tried to optimize its performance (reduce size of action head, hyperparameter optimization of training steps, LR, etc.). This helps to contextualize the results. Other papers have also used (vision-based!) DP with low numbers of demonstrations and achieved non-zero performance, e.g. https://arxiv.org/pdf/2403.19578 or https://arxiv.org/pdf/2501.14400.

---

### Official Review · Reviewer_TwmJ · 2026-05-12
**Review of Context-Adaptive Policy Framework for Robust and Reactive Robotic Manipulation vis Uncertainty-Aware Imitation Learning**

**Rating:** 8
**Confidence:** 3

**Review:**

This paper essentially proposes training three separate policies: the normal behavior cloning network (LfD policy), a stabilizing policy, and a goal attractor policy. The LfD policy is responsible for imitating expert actions when in distribution; the stabilizing policy takes the gradient with respect to the epistemic uncertainty of the data, thus creating attraction towards in distribution states; finally, the goal attract policy takes gradient with respect to the goal state belonging to the trajectory containing the state closest to the current state. The method decides which expert to take over depending on distance from expert demonstrations and from goal.

Strengths:
* The paper a well-motivated design, and the algorithm is grounded in defendable theory. It seeks to address the issue of distribution shift common in imitation learning.
* Good experiment results show promising improvement over base diffusion policy.
* Figures are interpretable and do well in getting point across. Writing is generally clear from any typos or grammatical errors.

Weaknesses:
* I am not convinced this method scales well to higher dimensional spaces. In high-dimensional spaces, such as images, whole-body manipulation, or dexterous hands, the curse of dimensionality kicks in, and I would suspect the kernels degrade in effectiveness, as everything looks equally far.
* Relatedly, mere "attraction" towards a the dataset could be ill-defined or a poor objective in several contact-rich manipulation tasks, where the presence of obstacles, or unilateral/asymmetric distances in task space (such as pushT tasks), makes the "shortest path" or a simple "attraction basin" less optimal/infeasible.
* I have concerns about the parameters in the algorithm that need tuning: kernel length scales, covariance scaling, position versus orientation weighting, stabilizer gains, goal attractor gains, etc.


Overall, I believe the method is interpretable, modular, and well supported by both LASA and real robot experiments. The real robot demonstrations are a notable strength and show meaningful improvements over individual policy components and diffusion policy baselines. However, the method’s stability claims should be weakened or formalized, and scalability remains limited due to the reliance on GPR and hand designed low dimensional task parameters. Overall, this is a useful and well executed contribution for robust task space LfD, though not a general solution for high dimensional or contact rich robot learning.

---

### Decision · Program_Chairs · 2026-05-21

Accept